# Evaluation of Healthcare Students’ Knowledge on Antibiotic Use, Antimicrobial Resistance and Antimicrobial Stewardship Programs and Associated Factors in a Tertiary University in Ghana: Findings and Implications

**DOI:** 10.3390/antibiotics11121679

**Published:** 2022-11-22

**Authors:** Israel Abebrese Sefah, Emmanuel Akwaboah, Emmanuel Sarkodie, Brian Godman, Johanna Caterina Meyer

**Affiliations:** 1Pharmacy Practice Department, School of Pharmacy, University of Health and Allied Sciences, Ho PMB 31, Ghana; 2School of Pharmacy, University of Health and Allied Sciences, Ho PMB 31, Ghana; 3University Hospital, Kwame Nkrumah University of Science and Technology, Kumasi PMB UPO KNUST, Ghana; 4Department of Public Health Pharmacy and Management, School of Pharmacy, Sefako Makgatho Health Sciences University, Pretoria 0204, South Africa; 5Strathclyde Institute of Pharmacy and Biomedical Sciences, University of Strathclyde, Glasgow G4 0RE, UK; 6Centre of Medical and Bio-allied Health Sciences Research, Ajman University, Ajman 346, United Arab Emirates

**Keywords:** antibiotics, antimicrobial resistance, antimicrobial stewardship, COVID-19, education, Ghana, healthcare students

## Abstract

Antimicrobial resistance (AMR) is a major public health problem globally, and Ghana is no exception. Good knowledge regarding antibiotic use, AMR, and the concept of antimicrobial stewardship (AMS) is critical among healthcare students to curb rising AMR rates in the future. Consequently, a need to ascertain this. A cross-sectional survey was undertaken among fifth-year pharmacy, medical students and fourth (final)-year nursing and physician assistantship students at the University of Health and Allied Sciences in Ghana to assess their knowledge on antibiotic use, AMR and AMS using a web-based self-administered structured questionnaire. Descriptive statistics, Fishers’ exact test, and multiple logistic regression analyses were performed. A total of 160 healthcare students were interviewed, of which 56.3% (*n* = 90) were male and 58.8% (*n* = 94) were in their fourth year of study. Good knowledge of antibiotic use, AMR, and AMS was associated with the study course (*p* = 0.001) and the number of years of study (*p* < 0.001). Overall, there were differences in the level of knowledge of antibiotics among the different healthcare students and their years of study. Efforts must now be made to enhance the curricula to ensure an improved and uniform transfer of knowledge of antibiotics, AMR, and AMS among the different healthcare students to sustain the fight against AMR in Ghana given growing concerns.

## 1. Introduction

In 2019, there were an estimated 4.95 million deaths globally associated with antimicrobial resistance (AMR), with estimates that up to 10 million lives will be lost annually to AMR by 2050 if no appropriate activities are instigated by governments and other key stakeholders to slow its progression [1,2]. There are also appreciable costs associated with AMR, with the World Bank believing that costs globally could rise to over $1 trillion per year by 2050, equivalent to 3.8% or more of gross domestic product, if the current situation continues unabated [3,4]. This combination is a concern especially among countries in sub-Saharan Africa, which currently have the highest rates of AMR worldwide, and they are on the increase [2]. In addition, across Africa, infectious diseases continue to pose a significant threat to human existence [5,6].

Antibiotics are one of the most commonly prescribed and dispensed in Africa [7], with appreciable prescribing and dispensing for self-limiting conditions including acute respiratory tract infections (ARIs) [8,9,10,11,12]. There are also concerns with high rates of inappropriate prescribing of antibiotics in hospitals across countries, including African countries, exacerbated by the COVID-19 pandemic [13,14,15,16]. This is important with increased prescribing of antibiotics, especially during the recent pandemic, driving up AMR [17,18,19,20,21,22].

These concerns have resulted in a number of global, regional, and national activities, including the World Health Organization (WHO) promoting National Action Plans (NAPs) to reduce AMR [3,23,24,25,26]. One of the five strategies to reduce the impact of AMR in the Global Action Plan is the need to optimize antimicrobial use through the implementation of antimicrobial stewardship programs (ASPs) [27,28,29]. Ghana, similar to other African countries, has developed its NAP and is currently implementing a range of agreed activities [30,31,32], with the goal to improve future antibiotic utilisation, reduce possible side-effects from antibiotics as well as reduce AMR [18,19,33,34,35,36,37,38].

The effective implementation of ASPs in hospitals requires a multidisciplinary team. However, there are concerns with manpower and resource issues affecting the sustainability of interventions, and knowledge gaps among practitioners, in low- and middle-income countries (LMICs) [39,40,41,42,43]. This is starting to be addressed in Africa, and will continue given the high and rising rates of AMR particularly in sub-Saharan Africa as well as increasing use of ‘Watch’ and ‘Reserve’ antibiotics [2,14,35,37,38]. The knowledge of healthcare students regarding antibiotics and AMR is a key factor to improving antimicrobial utilisation as they will play an important role in the future in either the prescribing or dispensing of antimicrobials across sectors; alternatively, advising key stakeholders including physicians and patients regarding their use [8,44,45,46]. Increased awareness on AMR among healthcare students has been shown to be an effective approach to improving future practitioners’ prescribing behavior [47]. This is important as there have been concerns with knowledge regarding antibiotics and AMR among healthcare professionals in Africa and the implications thereof [41,48,49,50].

However, most studies particularly in LMICs that have evaluated the knowledge of healthcare students regarding antibiotics and AMR have typically been conducted separately among medical students [47,51,52,53,54,55,56], pharmacy students [57,58,59,60,61,62], nursing and paramedic students with only a limited number of studies conducted among a combination of healthcare students [48,63,64,65,66,67,68]. This is a concern as a good uniform knowledge regarding antibiotic use, AMR, and ASPs is essential to optimize the future use of antimicrobials with all key stakeholders involved, which is in line with the goals of the Ghanaian NAP and beyond [30,32]. This is particularly important in Ghana given concerns with current high rates of AMR, considerable and inappropriate prescribing and dispensing of antibiotics across all sectors often without a prescription, and issues with poor compliance with national guidelines [12,69,70,71,72,73,74,75]. Currently, few hospitals in Ghana have attempted establishing ASPs. This is beginning to change with help from the UK Fleming fund in collaboration with UK institutions as well as input from the WHO giving guidance on implementing ASPs with the help of a toolkit [76,77]. This builds on successful ASPs already being implemented across Africa [35,37,38], and is in line with the goals of the Ghanian NAP [30,32]. However, to date, the Ministry of Health in Ghana is yet to roll-out a full-scale, nationwide implementation of ASPs at different levels of the healthcare system. This a concern considering the integral role AMS plays in health system strengthening by ensuring the optimum use of antimicrobials as part of key measures to combat AMR [34,36,76]. Alongside this, we are aware that educational programs can be effective in raising knowledge regarding antibiotics and ASPs in Ghana, helping to improve future antimicrobial prescribing [68,77].

Currently, little is known about antibiotics, AMR, and ASPs among healthcare students in Ghana. Consequently, we believed there was a need to conduct a study at the University of Health and Allied Sciences in Ghana, which is the only health and allied sciences university in Ghana. The objective is to evaluate healthcare students’ knowledge regarding antibiotic use, AMR, and ASPs, with the findings subsequently to be used to update educational activities to address ongoing concerns in Ghana.

We are aware that there have been studies seeking to increase the prescribing of topical antibiotics for acute respiratory infections. However, the principal focus among healthcare professionals (HCPs) should be to reduce antimicrobial use for potentially self-limiting viral infections [8,78].Consequently, such studies were considered outside the scope of this paper. Similarly, we have not considered studies demonstrating inappropriate antibiotic prescribing among dentists and dental surgeons, combined with potential ways to address this including local administration of antibiotics [79,80], as we did not included dental students in our study. Finally, we have not commented on ways to increase the uptake of COVID-19 vaccines across Africa by healthcare professionals (HCPs), including Ghana, given current appreciable hesitancy to reduce future serious illness and hospitalization [81], thereby reducing inappropriate antibiotic prescribing, as this is again outside the scope of this paper.

## 2. Results

We will first document the characteristics of the respondents before assessing any association between their characteristics and knowledge of antibiotics, AMR and ASPs.

### 2.1. Characteristics of the Respondents

A total of 160 students from the University of Health and Allied Sciences comprising of 33.1% (*n* = 53), 25.6% (*n* = 41), 29.4% (*n* = 47), and 11.9% (*n* = 19) nursing, medicine, physician assistantship, and pharmacy students, respectively, were surveyed, giving a response rate of 100% among those approached. This high rate was achieved with the help of constant reminders to recruited students, including emails and WhatsApp contact.

From the interviews conducted, 56.3% (*n* = 90) were male, 91.9% (*n* = 147) were within the age group of 20–25 years, 58.8% (*n* = 94) were in their final fourth year, 79.4% (*n* = 127) had no training in antibiotics prior to entering the university, and 77.0% (*n* = 120) had no close friends or relatives working in health-related fields (Table 1).

### 2.2. Association between Students’ Characteristics and Their Knowledge of Antibiotic Use, AMR, ASP and Their Overall Knowledge of Antibiotics

The Fisher’s exact test of independence showed a statistically significant association between the knowledge of AMR and gender (*p* = 0.004), and knowledge of antibiotics and training on antibiotics prior to entering the university (*p* < 0.000). There were associations between students’ knowledge on antibiotics use, AMR, ASP, and the course of study and the year of study. Overall knowledge of antibiotics was also associated with students’ course of study (*p* = 0.001) and year of study (*p* < 0.001) (Table 2).

### 2.3. Multiple Logistic Regression between Students’ Characteristics and Their Overall Knowledge on Antibiotics

Healthcare students’ overall knowledge on antibiotics was approximately six times (OR = 5.84, CI 2.09–16.26) more likely to be good knowledge if they were in their fifth year in the SOP and SOM, than if they were in their fourth year in the SONAM and SAHS (Table 3).

## 3. Discussion

We believe this is the first study of its kind in Ghana involving both health and allied science students in a single university. Encouragingly, there was a good overall level of knowledge regarding antibiotics among most of the healthcare students at the University of Health and Allied Sciences in Ghana, which is similar to some studies conducted among students in LMICs [53,54,58,59]. However, this is different from other studies conducted among students in LMICs where there have been concerns with their knowledge [46,55,56,64,82,83].

As seen, good knowledge of antibiotic use, AMR, and ASPs was associated with the study course (*p* = 0.001) and the number of years of study (*p* < 0.0001). A greater proportion of healthcare students from the School of Pharmacy (94.7%) and the School of Medicine (87.2%) in their fifth year had good knowledge of these matters versus their counterparts from the School of Nursing and Midwifery (62.3%) and the School of Allied and Health Sciences (63.4%), who were in their final year, i.e., their fourth year. This could be due to differences in the structure and depth of the curriculum among these different healthcare students regarding these key subject areas. As a result, leading to a greater exposure to the principles of antimicrobial stewardship (AMS) among the former group compared to the latter, similar to other studies conducted among LMICs [54,65,82,84]. The healthcare students from the School of Pharmacy were also observed as having a slightly better overall knowledge with respect to these subject areas. The exact cause of the differences among them is an important area for future research going forward.

The poor knowledge regarding antibiotic use, AMR, and AMS among nursing and physician assistantship students compared to other members of the healthcare team is a concern. This needs to be actively addressed going forward as good knowledge of these subjects among the entire multidisciplinary HCP team is an essential step towards optimizing the future use of antimicrobials. Consequently, immediate steps need to be taken to address these knowledge gaps among HCP students. Essential steps include an urgent review and refinement of current curricula especially among nursing and physician students to include greater input on AMR and AMS principles. This is because they are key stakeholders in the prescription, administration, and counselling on antibiotic use across all sectors in Ghana. Furthermore, there must be efforts to instigate mandatory, continuous professional development programs surrounding AMR and ASPs among all HCP groups post-qualification as Ghana strives to achieve the goals of the NAP [31,32]. This should help with combating rising rates of AMR across Ghana [31,85]. This will increasingly include hybrid approaches to learning [86], which is even more important post COVID-19 with the high inappropriate use of antimicrobials in Ghana and across countries to prevent and treat patients with COVID-19 [15,17,87,88].

We are aware of a number of limitations with this study. Firstly, this study is limited by the small sample size, as well as the non-inclusion of other core members of the AMS teams including laboratory personnel trained in microbiology. Secondly, we conducted this study in only one university in Ghana, which may affect the internal and external validity of our findings. Lastly, we used a questionnaire derived from published studies combined with the considerable knowledge of the co-authors. However, this was not validated among our study population. This though is similar to numerous other studies undertaken by the co-authors across countries utilizing their considerable knowledge and experience in this area. Notwithstanding these concerns, we believe our findings are robust, providing guidance for future activities in the developing and refining of curricula in this University and across Ghana.

## 4. Materials and Methods

### 4.1. Study Site and Population

The University of Health and Allied Sciences located in Ho, Ghana, was established by an Act of Parliament of Ghana (Act 828, December 2011) as a public university to provide higher education in health sciences in Ghana. There are seven schools within the University. These include the School of Allied Health Sciences (SAHS), the School of Basic and Biomedical Sciences, the School of Medicine (SOM), the School of Nursing and Midwifery (SONAM), the School of Pharmacy (SOP), the School of Sports and Exercise Medicine, and the School of Public Health. While all the schools offer undergraduate and postgraduate programs, only the SOM and the SOP offer a six-year undergraduate professional doctorate program.

This study was conducted amongst four of the schools, namely the SOP, SOM, SONAM, and the SAHS. The target study population included fifth-year pharmacy and medicine students and fourth and final-year nursing and physician assistantship students. Fifth-year medical and pharmacy students were chosen as this is the final year for all classroom lectures and assessments. The sixth year is devoted mainly for clinical practice in hospitals outside the university campus, aimed at the acquisition of practical knowledge and skills, unlike their nursing and physician assistantship student counterparts who undertake similar practical training after completion of their four-year degree program.

### 4.2. Study Design

A cross-sectional study design was employed to evaluate the knowledge of antibiotic use, AMR, and ASP among pharmacy, medical, physician assistantship and nursing students in the University of Health and Allied Sciences.

### 4.3. Sample Size and Sampling Procedure

Based on a student population of 252, comprising of 30 pharmacy students, 73 medical students, 84 nursing students and 65 physician assistantship students, a minimum sample size of 153 students was calculated, using the Raosoft Inc. online sample size calculator (http://www.raosoft.com/samplesize.html (accessed on 10 May 2021)), assuming an expected frequency of 50% to yield the largest sample size, at 80% power and 95% confidence level. The sample size was increased to 160 to account for any incomplete data. Probability proportional sampling, based on the size of each school’s student population, was used to estimate the number of students from each school to be included in the final sample of 160 students. Simple random sampling, using a random number generator (https://www.random.org (accessed on 10 May 2021)) was used within each school to recruit the required number of students with the help of a class list obtained from the administration for the survey, which included 19 pharmacy students, 47 medical students, 53 nursing students, and 41 physician assistants.

### 4.4. Data Collection

A structured self-administered 35-item questionnaire was developed using online google forms for the survey, based on published literature of similar student studies including validated questionnaires, combined with the considerable knowledge and experience of the co-authors [54,58,60,66,89]. We have used this approach previously to investigate key issues and their implications across LMICs [8,31,81,86,90,91,92,93]. The questionnaire included firstly the socio-demographic characteristics of participating students, followed by questions aimed at assessing their knowledge about antibiotic use, antibiotic resistance, and ASPs, using three response options, namely ‘Agree’, ‘Disagree’, and ‘Do not know’ (Appendix A). This included 7 questions assessing students’ knowledge regarding antibiotics, 12 questions on AMR, and 8 questions on ASPs.

The link for access to the questionnaire and the consent form was sent to the recruited students via both their collected e-mail addresses and WhatsApp contact numbers since these media are widely used by the students. The same questionnaire was answered by all the different categories of students who participated in the study since we wanted to assess and compare their basic knowledge on these important subject areas.

Data were collected between June 2021 and October 2021. All students who were sampled and consented to participate in the study were emailed the web-based designed questionnaire.

### 4.5. Data Analysis

Data generated from the completed online google forms in a Microsoft Excel format were imported into STATA version 14 (StrataCorp, College Station, TX, USA) for analysis. A total score was calculated for knowledge regarding antibiotic use, AMR, and ASPs, and dichotomized as a “good” versus “poor” score. A correct response was assigned a score of one while an incorrect response was assigned a zero score.

A total score ≥ 60% for knowledge on antibiotic use, AMR, and ASP was considered as good based on similar studies [46,60,62]. We are aware other studies have used lower and higher cut-off scores up to 80% for good knowledge [53,94,95]. However, we chose ≥ 60% based on previous published studies [46,60,62]. An overall knowledge score was subsequently determined using the same cut off score. Descriptive statistics were used to summaries variables using percentages for all categorical variables. Inferential statistics, using the Fisher’s exact test of independence, and multiple logistic regression using explanatory variables that were statistically significant after the former analysis were subsequently conducted to determine associated variables and predictors of overall knowledge of antibiotics respectively.

## 5. Conclusions

There are disparities in the overall level of knowledge of antibiotics, AMR, and ASPs among the different healthcare students at this University in Ghana. Efforts must be made to address these concerns in updated curricula as well as the continual development post-qualification. The objective is to ensure an improved and uniform transfer of knowledge on these subjects among the different student populations. This is imperative to sustain the fight against AMR in Ghana in line with the objectives of the NAP. We will be following this up in future studies.

## Figures and Tables

**Table 1 antibiotics-11-01679-t001:** Socio-demographic characteristics of respondents (*n*= 160).

Variable	Frequency (*n*)	Percentage (%)
Age (years) (*n* = 160)		
20–25	147	91.9
26–31	10	6.3
>31	3	1.9
Gender (*n* = 160)		
Male	90	56.3
Female	70	43.8
Course of study (*n* = 160)		
Medicine	47	29.4
Physician assistantship	41	25.6
Nursing	53	33.1
Doctor of pharmacy	19	11.9
Year of study (*n* = 160)		
Fourth year	94	58.8
Fifth year	66	41.3
Close friend/relation working in a health-related field (*n* = 160)		
Yes	120	75.0
No	40	25.0
Exposed to any antibiotic training before university (*n* = 160)		
Yes	33	20.6
No	127	79.4

**Table 2 antibiotics-11-01679-t002:** Association between socio-demographic characteristics of respondents and their knowledge of antibiotic use, resistance, stewardship and their overall knowledge of antibiotics.

Variable	Antibiotic Use *n* (%)		Antibiotic Resistance *n* (%)		Antibiotic Stewardship Programs *n* (%)		Overall Level of Knowledge *n* (%)	
	Good	Poor	Good	Poor	Good	Poor	Good	Poor
Age (N = 160)								
20–25 (*n* = 147)	115 (78.2)	32 (21.8)	83 (56.5)	64 (43.5)	112 (76.2)	35 (23.8)	109 (74.1)	38 (25.9)
26–31 (*n* = 10)	10	0	8	2	7	3	8	2
>31 (*n* = 3)	3	0	1 (33.3)	2 (66.7)	1 (33.3)	2 (66.7)	1 (33.3)	2 (66.7)
*Fisher’s Exact (p-value)*	0.294		0.264	0.221			0.253	
Gender (N = 160)								
Male (*n* = 90)	72	18	61 (67.8)	29 (32.2)	69 (76.7)	21 (23.3)	69 (76.7)	21 (23.3)
Female (*n* = 70)	56	10	31 (44.3)	39 (55.7)	51 (72.9)	19 (27.1)	49 (51.6)	21 (18.4)
*Fisher’s Exact (p-value)*	1.000		**0.004**		0.587		0.369	
Course of Study (N = 160)								
Medicine (*n* = 47)	47	0	31	16	37 (78.7)	10 (21.3)	41 (87.2)	6 (12.8)
Physician Assistantship (*n* = 41)	38 (92.7)	3 (7.3)	16 (23.6)	25	22 (53.7)	19 (46.3)	26 (63.4)	15 (36.6)
Nursing (*n* = 53)	24 (45.3)	29 (54.7)	27 (50.9)	26 (49.1)	43 (81.1)	10 (18.9)	33 (62.3)	20 (37.7)
Doctor of Pharmacy (*n* = 19)	19	0	18 (94.7)	1 (5.3)	18 (94.7)	1 (5.3)	18 (94.7)	1 (5.3)
*Fisher’s Exact (p-value)*	**0.000**		**0.000**		**0.002**		**0.001**	
Year of study (N = 160)								
Fourth year (*n* = 94)	62	32	43 (45.7)	51 (54.3)	65 (69.1)	29 (30.9)	59 (62.8)	35 (37.2)
Fifth year (*n* = 66)	66	0	49 (74.2)	17 (25.8)	55 (83.3)	11 (16.7)	59 (89.4)	7 (10.6)
*Fisher’s Exact (p-value)*	**0.000**		**0.000**		**0.044**		**0.000**	
Relative/friend working in a health-related field (N = 160)								
Yes (*n* = 120)	97 (80.8)	23 (19.2)	70 (58.3)	50 (41.7)	93 (77.5)	27 (22.5)	91 (75.8)	29 (24.2)
No (*n* = 40)	31 (77.5)	9 (22.5)	22	18	27 (67.5)	13 (32.5)	27 (67.5)	13 (32.5)
*Fisher’s Exact (p-value)*	0.653		0.716		0.213		0.306	
Exposure to antibiotic training before university (N = 160)								
Yes (*n* = 33)	33	0	18 (54.5)	15 (45.5)	22 (66.7)	11 (33.3)	26 (78.8)	7 (21.2)
No (*n* = 127)	95 (74.8)	32 (25.2)	74 (58.3)	53 (41.7)	98 (77.2)	29 (22.8)	92 (72.4)	35 (27.6)
*Fisher’s Exact (p-value)*	**0.000**		0.698		0.259		0.514	

NB: Emboldened *p*-value are those that are below the significance level of 0.05.

**Table 3 antibiotics-11-01679-t003:** Multiple logistic regression between students’ characteristics and their overall knowledge.

Variable	Adjusted Odds Ratio	95% Confidence Interval	*p*-Value
Age	0.4979	0.1746–1.4199	0.192
Gender	0.7114	0.3231–1.5661	0.398
Course of study	1.2495	0.7391–2.1120	0.406
Year of study	5.8428	2.0990–16.2637	**0.001**
Relative/friend working in a health-related field	0.7819	0.3386–1.8053	0.564
Exposure to antibiotic training before university	0.6686	0.250–1.7855	0.422

NB: The reference covariate used for the analysis of the year of study variable was the fourth year of study. Emboldened *p*-value is the one are those that are below the significance level of 0.05.

## Data Availability

Additional data are available from the corresponding authors on reasonable request.

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
