# Peer review of "Evaluation of Healthcare Students’ Knowledge on Antibiotic Use, Antimicrobial Resistance and Antimicrobial Stewardship Programs and Associated Factors in a Tertiary University in Ghana: Findings and Implications"

_antibiotics, 2022, doi:10.3390/antibiotics11121679_

Round 1

Reviewer 1 Report

Abstract-Overall structure to be improved by including more highlights about the methods and results and to cut make the introduction part more precise.

Introduction

The relevance of the study with regard to the current AMR rates in ghana has been clearly stated.However a better overview of the ASPs prevalent in Ghana and the current place of educational programmes focusing on the topics of concern would have been better touched upon.

Results 

Scientific language should be reinforced.

The method of sampling is not specified.

The type of interview and how the questionnaires were distributed among the participants have to be specified.Was is a qualitative interview?Were the interviews done online or as soft copy form?The questions used to evaluate each key concept have to be enlisted .Was the questionnaire a validated one?Was the same questionnaire used for medical students ,pharmacy ,nursing and allied health care students?

The variable -year of study in Table 1 lacks clarity.What is the unit used for year of study.What does 400 and 500 mean?

Discussion

The way in which the results of the study are potrayed as opportunities for improvement in improving prescription appropriateness is quite confusing.The authors should disccuss possible causes of reasonably higher level of knowledge in the interviewed population.As the interviwed population is widely heterogenous in their existing curriculum the differences in their knowledge which is elicited by same questinnaire is not commendable.

Author Response

Comment: Abstract-Overall structure to be improved by including more highlights about the methods and results and to cut make the introduction part more precise.

Author comments: Thank you for this. The abstract has now been updated within the confines of the Journal. We hope this is now OK.

Comment: Introduction - The relevance of the study with regard to the current AMR rates in Ghana has been clearly stated. However a better overview of the ASPs prevalent in Ghana and the current place of educational programmes focusing on the topics of concern would have been better touched upon.

Author comments: Thank you for this comment. Extra details of ASPs in Ghana have now been added to the introduction as suggested – building on recent successful ASPs throughout Africa (one new review included). We hope this is n ow OK.

Comment: Results - Scientific language should be reinforced. The method of sampling is not specified.

The type of interview and how the questionnaires were distributed among the participants have to be specified. Was is a qualitative interview? Were the interviews done online or as soft copy form? The questions used to evaluate each key concept have to be enlisted .Was the questionnaire a validated one? Was the same questionnaire used for medical students, pharmacy, nursing and allied health care students?

Author comments: Thank you for this comments and suggestion. We have now upgraded the results. In fact, the sampling method for the study was simple random sampling and has been stated under the heading “Sample size and sampling procedure” in the method section (4.3). The type of interview and method of distribution of questionnaire to participants have now been elaborated in the data collection section. The method helped to ensure a high response rate.

We have also clarified in the data collection section that the same questionnaire was used to assess and compare their knowledge on these subjects’ areas (4.4). We have stated that the questionnaire was adapted from previous studies combined with the considerable knowledge and experience of the co-authors (References provided). However, it was not validated which has been stated as a limitation in the study. We hope this is now OK.

Comment: The variable -year of study in Table 1 lacks clarity. What is the unit used for year of study. What does 400 and 500 mean?

Author comment: Thank you for this. The categorization of the variable ‘year of study’ was now been clarified as fourth and fifth year of study. We hope this is okay now.

Comment: Discussion - The way in which the results of the study are portrayed as opportunities for improvement in improving prescription appropriateness is quite confusing. The authors should discuss possible causes of reasonably higher level of knowledge in the interviewed population. As the interviewed population is widely heterogeneous in their existing curriculum the differences in their knowledge which is elicited by same questionnaire is not commendable.

Author comment: Thank you for this. We have now expanded on the possible causes of the knowledge gap among the different healthcare students and ways forward. We hope this is now acceptable.

Reviewer 2 Report

A brief summary 

The objective of this study was to evaluate healthcare students’ knowledge regarding antibiotic use, antimicrobial resistance (AMR) and antimicrobial stewardship programs (ASPs).

The strength of the analysis could be that this was the first study in Ghana and involving both health and allied science students in a single university.

General notes:

This is an original article and contains 93 references. This is a lot, during the review period (5 days) it is not possible to review/check these 93 references. For the revision, I ask you to significantly reduce the number of references. Some of the references seem to be not necessary / marginal, but I couldn't go through all of them. For example, reference 22 related to a special situation.

I am missing a table where the distribution of the answers to the questions of the questionnaire (questions 9-35) can be seen item by item. For example, there is a similar table in reference 65 and 51

Self- citations:  Of the 93 references: Brian Godman's 7 articles (9,12,16,18, 33, 90, 93), and Israel Abebrese Sefah's article: 5 items (48, 74, 75, 87). In summary, 11 articles are self-citations, if I counted correctly.

General concept comments

1.     Is the manuscript clear, relevant for the field and presented in a well-structured manner?

Partly. I have proposed significant changes below.

  1. Are the cited references current (mostly within the last 5 years)?

I didn't have time to check, there are 93 references in the article. Based on an estimate, most of them have recent (within the last 5 years) references.

3.     Does it include an abnormal number of self-citations?

I don't know, there are a total of 11 self-citations, which is 11%, I think this is more than usual.

4.     Is the manuscript scientifically sound and is the experimental design appropriate to test the hypothesis?

Mainly. I have proposed minor changes below.

5.     Are the manuscript’s results reproducible based on the details given in the methods section?

Yes.

6.     Are the figures/tables/images/schemes appropriate? Do they properly show the data? Are they easy to interpret and understand? Are the data interpreted appropriately and consistently throughout the manuscript? Please include details regarding the statistical analysis or data acquired from specific databases.

Yes, the tables are correct, I have only a minor notes:

Table1. “Year of study” variables: What do “400” and “500” mean?

Table 2. Editing suggestion: Use the same style in the table, i.e. the titles should be either in the middle (e.g. Gender) or on the left side (e.g. Year of study). The vertical grids are there in the lower part of the table, while they are missing in the upper part. Have a uniform appearance.

Table 3. ODDS Ration, and 95%CI: No need for 4 decimal places ; What were the base line of the variable “Year of the study” and “Gender” ?

7.     Are the conclusions consistent with the evidence and arguments presented?

Yes.

8.     Please evaluate the ethics statements and data availability statements to ensure they are adequate.

 Yes, it is adequate.

Specific comments:

Abstract: I have no suggestion

Introduction

The introduction is long (1 page) and contains 73 references, which is lot. I recommend shortening.

Methods:

Was the questionnaire validated? Was Cronbach's alpha calculated?

Line 232:

“A total score ≥ 60% for knowledge on antibiotic use, AMR and ASP were considered as good based on similar studies [51,65,67].”

60% is a very low limit considering that the questions are simple.

Reference 67 (Pharm Pract (Granada). 2015 Jan-Mar; 13(1): 523.): I did not find this (60%) cut off value in the article. In this article - as I saw it - the analysis is done as a continuous variable.

Reference 65 (Curr Pharm Teach Learn. 2017;9(4):551-9.) I did not find this (60%) cut off value in the article.

Reference 51 (PLoS One. 2021;16(5):e0251301.) A pass mark of 60% was considered, in this article. This study use this cut off value (60 % )

Can knowledge above 60% really be considered good and adequate? What do the authors think about this? Maybe there is literature in which a higher cut off value was used? What confirms the use of this value? Why was the “knowledge %” not analysed as a continuous variable?

Result

Line 103: “ ..giving a response rate of 100% among those approached. “

This is very exceptional, how did you achieve this?

Minor comment:

Line 251: “…IS, EA, ES, BG, JCM; visualization, “

There are no graphs or figures in the article. What does visualization mean?

Author Response

Comment: The objective of this study was to evaluate healthcare students’ knowledge regarding antibiotic use, antimicrobial resistance (AMR) and antimicrobial stewardship programs (ASPs).

The strength of the analysis could be that this was the first study in Ghana and involving both health and allied science students in a single university.

Author comment: Thank you for this comment – appreciated!

Comment: This is an original article and contains 93 references. This is a lot, during the review period (5 days) it is not possible to review/check these 93 references. For the revision, I ask you to significantly reduce the number of references. Some of the references seem to be not necessary / marginal, but I couldn't go through all of them. For example, reference 22 related to a special situation.

Author comment: Thank you for this. We realise that we had an appreciable number of references in the Introduction, and have re-looked at these and cut down where we can. However, other Reviewers have asked for additional references which we have now included. Overall:

  • The references in the first paragraph (Lines 41 to 50) demonstrate the morbidity, mortality and costs of AMR – which need to be urgently addressed
  • The references in the second paragraph (Lines 51 – 57) discuss the appreciable prescribing and dispensing of antibiotics across sectors enhancing AMR – made worse by the COVID-19 pandemic – hence the need to include these to lay the foundation for the study
  • Third/ Fourth paragraph – Lines 58 – 80 – A key response to this threat is the generation of NAPs (including Ghana) and ASPs. However – ASPs have been difficult to perform in LMICs due to resource/ personnel issues. This is changing though in Africa as seen by recent reviews (cited). Ways forward include increasing knowledge among students including greater consideration of the AWaRe classification due to high use of Watch antibiotics in sub-Saharan Africa (again key references cited)
  • Lines 81 – 103 – Key rationale for this paper including limited studies especially in other LMICs involving all key student groups. We felt we needed to include an appreciable number of references in order to make this point robustly and we hope you agree
  • Lines 110 – 120 – These considerations were included at the request of a reviewer.

Discussion – The additional references – including additional requests – have been carefully considered. This includes paragraph 1 showing differences/ similarities with other studies which is key to any Discussion. Hybrid learning has increased with the pandemic and is here to stay (referenced).

Methods: The last remaining references largely deal with why a 60% cut off and also considerable justification for the method chosen. We felt we again needed to justify out approach given some of the comments from the other Reviewers. We trust you agree.

Comment: I am missing a table where the distribution of the answers to the questions of the questionnaire (questions 9-35) can be seen item by item. For example, there is a similar table in reference 65 and 51

Author comment: Thank you for this. In line with other Reviewers – we have chosen to consolidate the findings to enhance understanding of the key findings. We trust this is acceptable.

Comment - Self- citations:  Of the 93 references: Brian Godman's 7 articles (9,12,16,18, 33, 90, 93), and Israel Abebrese Sefah's article: 5 items (48, 74, 75, 87). In summary, 11 articles are self-citations, if I counted correctly.

Author comment: Thank you for this. We have removed some references but kept a number of others as we believe they are essential for this paper. We hope you agree. These include:

  • New Ref 8 (old 9) – showing that there are concerns across countries, including African countries, as excessive use of antibiotics for ARIs and the implications. The same reference has been used to justify why we did not look at topical antibiotics and why we feel justified in our approach (Methods section)
  • Old Ref 12 and 16 – Now removed
  • New ref 16 (Old 18) – Kept as a key paper demonstrating excessive use of antibiotics in patients with COVID-19 across countries despite limited bacterial or fungal infections – hence raising the spectre of AMR
  • New 31 (old 33) – Essential as this demonstrates widespread activities across Africa including Ghana regarding the development and implementation of NAPs
  • New 69/ 70 (old 74,75) – Essential as demonstrate concerns with current antimicrobial prescribing in Ghana (along with others)
  • New 77 (old 48) – Essential to help demonstrate limited ASP activities in Ghana until recently with initiatives such as this
  • Ref 87 – Essential as shows high rates of inappropriate use of antimicrobials (hydroxychloroquine antibiotics) among patients with COVID-19 in ambulatory care in Ghana despite limited/ no evidence of their value
  • Methods section – As stated earlier – we believe we had to include a number of self-citations here to justify our approach (along with other references)

We have in fact recently published on SAP in Ghana - Sefah IA, Denoo EY, Bangalee V, Kurdi A, Sneddon J, Godman B. Appropriateness of surgical antimicrobial prophylaxis in a teaching hospital in Ghana: findings and implications. JAC Antimicrob Resist. 2022;4(5):dlac102, as well as on ART - Sefah IA, Mensah F, Kurdi A, Godman B. Barriers and facilitators of adherence to antiretroviral treatment at a public health facility in Ghana: a mixed method study. Hospital Practice. 2022;50(2):110-7.  However, we decided not to include these references in view of concerns with excessive citations. We hope this is now acceptable.

Comment:   Is the manuscript clear, relevant for the field and presented in a well-structured manner? Partly. I have proposed significant changes below.

Author comment: Thank you for this. Suggested changes by the reviewer have been made and issues raised addressed. We hope these are acceptable.

Comment: Are the cited references current (mostly within the last 5 years)?

I didn't have time to check, there are 93 references in the article. Based on an estimate, most of them have recent (within the last 5 years) references.

Author comment: Thank you – yes most are current to help justify this paper

Comment:   Does it include an abnormal number of self-citations?

I don't know, there are a total of 11 self-citations, which is 11%, I think this is more than usual.

Author comment: Thank for this. As seen above we have removed some references where not seen as essential to help justify this paper/ our approach going forward (and not included others). We hope you agree.

Comment:  Is the manuscript scientifically sound and is the experimental design appropriate to test the hypothesis? Mainly. I have proposed minor changes below.

Author comment: Thank you for this comment. We hope we have successfully addressed the issues raised.

Comment: Are the manuscript’s results reproducible based on the details given in the methods section?

Yes.

Author comment: Thank you for this comment.

Comment: Are the figures/tables/images/schemes appropriate? Do they properly show the data? Are they easy to interpret and understand? Are the data interpreted appropriately and consistently throughout the manuscript? Please include details regarding the statistical analysis or data acquired from specific databases. Yes, the tables are correct, I have only a minor notes:

Author comment: Thank you for this. Issues raised on the table has now been addressed. We hope it is okay now.

Comment: Table1. “Year of study” variables: What do “400” and “500” mean?

Author comment: Thank you for this. This variable in the table has now been clarified. The categorization under this variable has been changed from levels 400 and 500 to fourth and fifth year of study respectively, and hope this is now OK.

Comment: Table 2. Editing suggestion: Use the same style in the table, i.e. the titles should be either in the middle (e.g. Gender) or on the left side (e.g. Year of study). The vertical grids are there in the lower part of the table, while they are missing in the upper part. Have a uniform appearance.

Author comment: Thank you for this. The same style in all the tables have now been ensured. We have now pushed the headings to the left hand side

Comment: Table 3. ODDS Ration, and 95%CI: No need for 4 decimal places ; What were the base line of the variable “Year of the study” and “Gender” ?

Author comment: Thank you for this. We have reduced the decimal places in Table 3 to two, and hope this is now OK. Since only the year of study show statistical significance, we have indicated the reference co-variate under the table to help with interpretation, and how this is acceptable.

Comment: Are the conclusions consistent with the evidence and arguments presented? Yes.

Author comment: Thank you for this.

Comment: Please evaluate the ethics statements and data availability statements to ensure they are adequate. Yes, it is adequate.

Author comment: Thank you for this evaluation

Comment: Abstract: I have no suggestion

Author comment: Thank you for this.

Comment: Introduction: The introduction is long (1 page) and contains 73 references, which is lot. I recommend shortening.

Response to comment: Thank you for this. As seen – we have made some changes. However kept others to justify our approach (including addressing concerns from the other Reviewers). We hope this is OK.

Comment:  Was the questionnaire validated? Was Cronbach's alpha calculated?

Response to comment: Thank you for this. We did not validate the questionnaire among our study population and this limitation as now been stated in the paper. However, we have provided a number of references justifying our approach, and hope this is now OK.

Comment: Line 232: “A total score ≥ 60% for knowledge on antibiotic use, AMR and ASP were considered as good based on similar studies [51,65,67]. 60% is a very low limit considering that the questions are simple. Reference 67 (Pharm Pract (Granada). 2015 Jan-Mar; 13(1): 523.): I did not find this (60%) cut off value in the article. In this article - as I saw it - the analysis is done as a continuous variable. Reference 65 (Curr Pharm Teach Learn. 2017;9(4):551-9.) I did not find this (60%) cut off value in the article. Reference 51 (PLoS One. 2021;16(5):e0251301.) A pass mark of 60% was considered, in this article. This study use this cut off value (60 % ). Can knowledge above 60% really be considered good and adequate? What do the authors think about this? Maybe there is literature in which a higher cut off value was used? What confirms the use of this value? Why was the “knowledge %” not analyzed as a continuous variable?

Author comment: Thank you for this comment.  We have removed reference number 65 as it was wrongly cited and has replaced this with: Asogwa I, Offor S, Mbagwu H. Knowledge, attitude and practice towards antibiotics use among non-medical university students in Uyo, Nigeria. J Adv Med Pharm Sci. 2017;15(1):1-1. We choose the 60% cut off point among other cut off points used in the studies as this is the average cut off point used in most such studies (quoted). To help further justify this – we have included studies that have used higher and lower cut-off levels, and hope this is now acceptable.

Comment: Line 103: “ ..giving a response rate of 100% among those approached. “ This is very exceptional, how did you achieve this?

Author comment: Thank you for this comment.  We achieved this due to frequent reminders of the recruited student via their email and whatsapp contact numbers – now inserted

Comment: Line 251: “…IS, EA, ES, BG, JCM; visualization. There are no graphs or figures in the article. What does visualization mean?

Author comment: Thank you for this. We have removed the visualization aspect of authors’ contribution.

Reviewer 3 Report

Many thanks for this submission: the topic is Interesting and innovative, ho weber some modifications are needed.

line 55: authors should consider the topical administration of antibiotics and the global intention to reduce antibiotics prescription: please insert this phrase Ar Line 55:

”..For this reason some authors introduced a possible use of topical antibiotics, ehi le Many articles underlined a possible global reduction of antibiotics administration..”

please cite the following:

Essack S, Bell J, Burgoyne DS, Duerden M, Shephard A. Topical (local) antibiotics for respiratory infections with sore throat: An antibiotic stewardship perspective. J Clin Pharm Ther. 2019 Dec;44(6):829-837. doi: 10.1111/jcpt.13012. Epub 2019 Aug 13. PMID: 31407824; PMCID: PMC6899613.

Busa A, Parrini S, Chisci G, Pozzi T, Burgassi S, Capuano A. Local versus systemic antibiotics effectiveness: a comparative study of postoperative oral disability in lower third molar surgery. J Craniofac Surg. Essack S, Bell J, 

D'Ambrosio F, Di Spirito F, De Caro F, Lanza A, Passarella D, Sbordone L. Adherence to Antibiotic Prescription of Dental Patients: The Other Side of the Antimicrobial Resistance. Healthcare (Basel). 2022 Aug 27;10(9):1636. doi: 10.3390/healthcare10091636. PMID: 36141247; PMCID: PMC9498878.

Chisci G, Hatia A. Antibiotics in orthognathic surgery and postoperative infections. Int J Oral Maxillofac Surg. 2022 Aug 27:S0901-5027(22)00320-4. doi: 10.1016/j.ijom.2022.08.008. Epub ahead of print. PMID: 36041952.

reference 94 not present.

Author Response

Comment: Many thanks for this submission: the topic is Interesting and innovative, though some modifications are needed.

Author comment: Thank you for the encouraging comment

Comment: line 55: authors should consider the topical administration of antibiotics and the global intention to reduce antibiotics prescription: please insert this phrase Ar Line 55:

Author comment: Thank you – we have now added comments about topical administration at the end of the Introduction and stated that outside the scope of this general paper. We hope you agree.

Comment:”..For this reason some authors introduced a possible use of topical antibiotics, ehi le Many articles underlined a possible global reduction of antibiotics administration..”

please cite the following:

Essack S, Bell J, Burgoyne DS, Duerden M, Shephard A. Topical (local) antibiotics for respiratory infections with sore throat: An antibiotic stewardship perspective. J Clin Pharm Ther. 2019 Dec;44(6):829-837. doi: 10.1111/jcpt.13012. Epub 2019 Aug 13. PMID: 31407824; PMCID: PMC6899613.

Busa A, Parrini S, Chisci G, Pozzi T, Burgassi S, Capuano A. Local versus systemic antibiotics effectiveness: a comparative study of postoperative oral disability in lower third molar surgery. J Craniofac Surg. 

Essack S, Bell J, D'Ambrosio F, Di Spirito F, De Caro F, Lanza A, Passarella D, Sbordone L. Adherence to Antibiotic Prescription of Dental Patients: The Other Side of the Antimicrobial Resistance. Healthcare. 2022 Aug 27;10(9):1636. doi: 10.3390/healthcare10091636. PMID: 36141247; PMCID: PMC9498878.

Chisci G, Hatia A. Antibiotics in orthognathic surgery and postoperative infections. Int J Oral Maxillofac Surg. 2022 Aug 27:S0901-5027(22)00320-4. doi: 10.1016/j.ijom.2022.08.008. Epub ahead of print. PMID: 36041952.

Author comment: Thank you for this. As stated above – we have included a number of these papers but also stated why these issues are outside the scope of this paper – especially as we did not include dental students in our sample. This was a deliberate decision initially as some of the co-authors have also published on inappropriate prescribing of antibiotics among dentists - Fadare JO, Oshikoya KA et al. Patterns of drugs prescribed for dental outpatients in Nigeria: findings and implications. Acta Odontol Scand. 2017;75(7):496-506 – however now amended. Finally, we did not include the reference to patients (Antibiotics paper) as this was aimed at patients and not students. We hope this is acceptable.

Comment: reference 94 not present.

Author comment: Thank you for this. The number 94 has now been deleted.

Round 2

Reviewer 3 Report

Accept